# Carotid plaque thickness is increased in chronic kidney disease and associated with carotid and coronary calcification

**Sasha S. Bjergfelt[1], Ida M. H. Sørensen[1], Henrik Ø. Hjortkjær[2], Nino Landler[3], Ellen L. F. Ballegaard[1], Tor Biering-Sørensen[3], Klaus F. Kofoed[2], Theis Lange[4], Bo Feldt-Rasmussen[1], Henrik Sillesen[5], Christina Christoffersen[6,7], Susanne Bro[1]***

1 Department of Nephrology, Rigshospitalet, University of Copenhagen, Copenhagen, Denmark,
2 Department of Cardiology, Rigshospitalet, University of Copenhagen, Copenhagen, Denmark,
3 Department of Cardiology, Herlev-Gentofte Hospital, University of Copenhagen, Copenhagen, Denmark,
4 Department of Public Health (Biostatistics), University of Copenhagen, Copenhagen, Denmark,
5 Department of Vascular Surgery, Rigshospitalet, University of Copenhagen, Copenhagen, Denmark,
6 Department of Clinical Biochemistry, Rigshospitalet, University of Copenhagen, Copenhagen, Denmark,
7 Department of Biomedical Sciences, University of Copenhagen, Copenhagen, Denmark

* susanne.bro@regionh.dk

**Data Availability Statement:** The datasets generated during the current study are not publicly available due to Danish legal restrictions but are

## Abstract

### Background

Chronic kidney disease accelerates both atherosclerosis and arterial calcification. The aim of the present study was to explore whether maximal carotid plaque thickness (cPTmax) was increased in patients with chronic kidney disease compared to controls and associated with cardiovascular disease and severity of calcification in the carotid and coronary arteries.

### Methods

The study group consisted of 200 patients with chronic kidney disease stage 3 from the Copenhagen Chronic Kidney Disease Cohort and 121 age- and sex-matched controls. cPTmax was assessed by ultrasound and arterial calcification by computed tomography scanning.

### Results

Carotid plaques were present in 58% of patients (n = 115) compared with 40% of controls (n = 48), p = 0.002. Among participants with plaques, cPTmax (median, interquartile range) was significantly higher in patients compared with controls (1.9 (1.4–2.3) versus 1.5 (1.2–1.8) mm), p = 0.001. Cardiovascular disease was present in 9% of patients without plaques (n = 85), 23% of patients with cPTmax 1.0–1.9 mm (n = 69) and 35% of patients with cPTmax >1.9 mm (n = 46), p = 0.001. Carotid and coronary calcium scores >400 were present in 0% and 4%, respectively, of patients with no carotid plaques, in 19% and 24% of patients with cPTmax 1.0–1.9 mm, and in 48% and 53% of patients with cPTmax >1.9 mm, p<0.001.

available on reasonable request from The Steering committee of the CPH CKD cohort project (secretary Christine.Korsholm.Nielsen@regionh.dk), provided relevant ethical and legal permissions have been attained priorly and researchers meet the criteria for access to confidential data.

**Funding:** This study was supported by Rigshospitalets Forskningspulje, https://www.forskningspuljer-rh.dk/ (SSB) and the Augustinus Foundation, https://augustinusfonden.dk/, grant no. 19-2323 (BFR). The funders had no role in study design, data collection and analysis, decision to publish, or preparation of the manuscript.

**Competing interests:** I have read the journal´s policy and the authors of this manuscript have the following competing interests: BFR reports research grants from The NovoNordisk Foundation (Steno Collaborative Grant), HS reports research grants from Philips Ultrasound and Bayer and honoraria from Bayer, Novo Nordisk, Bracco and Philips Ultrasound, TBS reports research grants from Sanofi Pasteur and GE Healthcare, the Lundbeck Foundation and the Novo Nordisk Foundation during the conduct of the study. All other authors: no competing interests.

## Conclusions

This is the first study showing that cPTmax is increased in patients with chronic kidney disease stage 3 compared to controls and closely associated with prevalent cardiovascular disease and severity of calcification in both the carotid and coronary arteries.

## Introduction

Chronic kidney disease (CKD) is associated with an increased risk of cardiovascular disease [1–5]. Indeed, patients with CKD are far more likely to die from cardiovascular disease than progress to end stage kidney disease requiring dialysis or kidney transplantation [6].

Early diagnosis of cardiovascular disease in patients with CKD is challenging, since these patients do not demonstrate typical clinical symptoms [7]. Moreover, prediction of future cardiovascular disease based on classical risk scoring systems underestimates the risk in patients with CKD [8]. An alternative approach for predicting symptomatic cardiovascular disease is based on the assessment of asymptomatic vascular disease using coronary artery calcium score (CACS) and carotid ultrasound imaging techniques. CACS seems to predict future cardiovascular disease for the individual person in the general population much better than risk-factor based scoring systems [9, 10]. Quantification of plaque size from both carotids by ultrasound has a similar predictive value as CACS in the general population [11]. Accordingly, a recent study of 6102 asymptomatic (non-CKD) persons showed that measurement by ultrasound of the simpler maximal carotid plaque thickness (cPTmax) predicted cardiovascular events similarly to the more comprehensive carotid plaque burden estimates [12].

Also in CKD, CACS provides additional value beyond existing clinical risk factor scoring systems [13–16]. So far, only few prospective ultrasound studies of the carotid artery have been performed in the CKD setting, and they used different methods for assessment of carotid atherosclerosis [17, 18].

The present study is the first to evaluate cPTmax in patients with CKD. The primary aim was to examine whether cPTmax is increased in patients with CKD stage 3 compared with controls and associated with prevalent cardiovascular disease. The secondary aim was to explore the association between cPTmax and the severity of calcification in the carotid and coronary arteries in CKD patients.

## Materials and methods

### Study population

The present carotid ultrasound study was a cross-sectional baseline study of a sub-group of patients from the Copenhagen (CPH) CKD Cohort described in details elsewhere [19]. In brief, the CPH CKD Cohort is a single-centre prospective, observational study examining novel risk factors and imaging methods for early detection of cardiovascular disease in a cohort of patients aged 30–75 years with any diagnosis of CKD stages 1–5 (no dialysis). CKD and CKD stages were defined in accordance with the 2012 Kidney Disease Improving Global Outcomes Guidelines [20]. The estimated glomerular filtration rate (eGFR) used to define eligibility was based on a measured plasma creatinine value and the CKD-EPI$_{krea}$ formula [21]. In total, 741 patients with CKD, 62 (8%) at stage 1, 115 (16%) at stage 2, 375 (51%) at stage 3, 146 (20%) at stage 4, and 43 (6%) at stage 5, were enrolled consecutively from the nephrology

outpatient clinic at Rigshospitalet, CPH University Hospital, Denmark, between 2015 and 2017.

Among included patients, 580 accepted to undergo computed tomography (CT) scanning of the major arterial regions: The carotid and coronary arteries, the thoracic and abdominal aorta and the iliac arteries [19]. The carotid ultrasound study was restricted to patients with CKD stage 3 (eGFR 30–59 ml/min per 1.73 m$^2$). This group was selected based on the hypothesis that identification of vascular disease in early CKD through screening programs may potentially maximize the effects of intervention to reduce progression of cardiovascular disease. Eligible patients from the CPH CKD Cohort were contacted and included consecutively until the number of 200 was reached.

Controls for the CPH CKD study were recruited by posting notices on the web-site "forsoegsperson.dk" and in local newspapers. The inclusion criteria were age 30–75 years, no history of cardiovascular disease, CKD, or other chronic or malignant disease. The exclusion criteria were eGFR <60 or kidney damage. The presence of well-treated mild hypertension (maximum 1 antihypertensive drug), thyroid disease and mild depression as well as hypercholesterolemia with or without statin treatment was accepted. From February to August 2017, we recruited a total of 175 controls who met the criteria and were sex- and age-matched at group level with the CPH CKD Cohort. Among these, a subgroup of 121 controls were selected to secure an optimal age and sex match at group level with the CKD patients undergoing ultrasound examination of the carotid arteries.

The study followed the principles of the Declaration of Helsinki and was approved by the Regional Scientific Ethical Committee (H-3-2011-069) and the Danish Data Protection Agency (30–0840). All participants signed a written informed consent at inclusion.

## Clinical data and biochemistry

Information on demographic and clinical data, medical history, medications and lifestyle factors were retrieved from in-person interviews, physical examination and electronic medical records [19]. Biochemical data were obtained from a fasting blood sample and a urine sample.

Cardiovascular disease was defined as a composite of prevalent coronary artery disease, a history with previous cerebrovascular infarction, carotid endarterectomy or stenting and/or peripheral artery disease as described in detail [19].

## Measurement of cPTmax

Ultrasound examination of the carotid arteries was performed by the same person (SSB) with a Philips EPIQ 7C ultrasound system equipped with a L12-3 transducer (48 Hz). The scanning protocol included a cross-sectional 10 second video of the carotid artery on both sides as described in detail [12]. The transducer was moved in the cranial direction from the proximal common carotid artery, just above the clavicle, to the bifurcation and then following the internal carotid artery as far as possible. All videos were made with the transducer in a perpendicular angle to the arteries. To support the assessment of the videos, longitudinally imaging of the common carotid artery and its branches were made. The assessment of cPTmax was made manually by a single reader (SSB) using a Dicom viewer (Micro Dicom). cPTmax was measured by finding the thickest part of the plaque in the cross-sectional videos and then measuring the radial distance from the media–adventitia interface to the intima–lumen interface towards the center of the arterial lumen [12] (Fig 1). Carotid plaque was defined as a focal structure encroaching into the arterial lumen of at least 0.5 mm, or 50% of the surrounding intima-media thickness value; or demonstrating a thickness ≥1.5 mm, as measured from the media-adventitia interface to the intima-lumen interface [22, 23]. For the statistical analysis

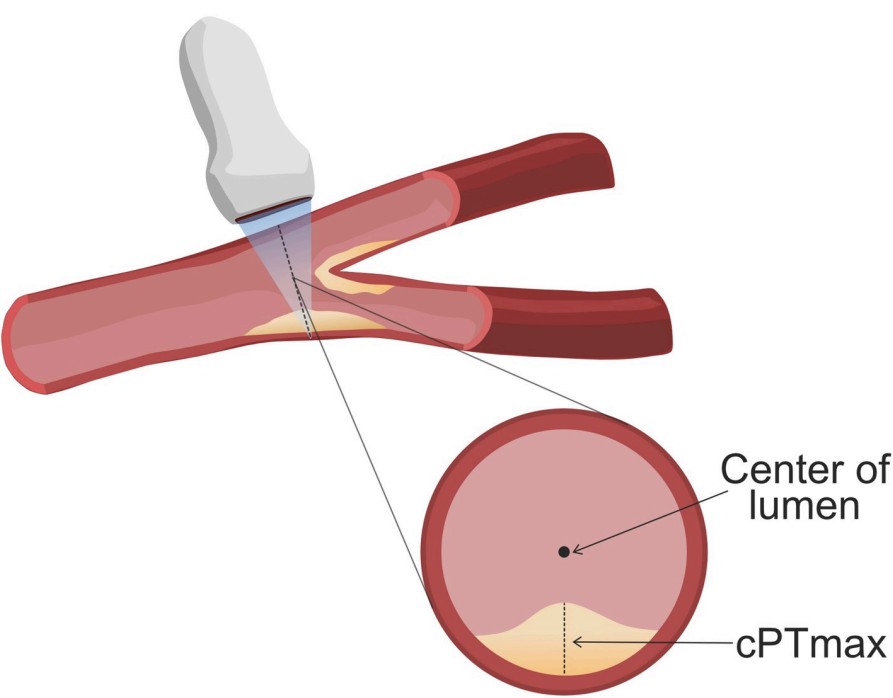

**Fig 1. Measurement of maximal carotid plaque thickness (cPTmax).** Image showing a segment of the carotid artery with a plaque, which is scanned with a transducer. The transverse section of the carotid artery shows how cPTmax is measured.

only the anatomical side with the highest cPTmax was used. Based on the distribution of cPTmax, the subjects were divided into 3 groups: No plaques, cPTmax 1.0–1.9 mm and cPTmax>1.9 mm (the median cPTmax in the CKD group was 1.9 mm). Intraobserver coefficient of variation (CV) was 9%. Re-examination of the videos for intra-observer variability testing (n = 30) did not re-classify any of the patients.

## Assessment of carotid and coronary artery calcium scores

Noncontrast electrocardiography-gated CT scans of the major arterial regions were performed with a 320-detector CT scanner (Aquillon One, Toshiba medical Systems, Japan). Calcification scoring was performed as previously described [19, 24]. Based on the distribution of Agatston scores in the selected arterial region, the subjects were divided into calcium score categories of 0 (no calcification), 1–100, 101–400, >400 [25]. Images of inadequate quality were excluded. Intra-observer CV was 10%, interobserver CVs varied between 12–13% [19].

## Statistical analyses

Statistical analyses were performed using SPSS version 25 (IBM SPSS Statistics, New York, USA). A value of $p<0.05$ was considered statistically significant. Categorical variables are presented as n (%) and were compared using the chi-square test. Totals may not add up to 100% due to rounding. Continuous variables were presented as means ± standard error of the means (SE) for normally distributed data and compared using the t-test, whereas skewed data were reported as median [interquartile range (IQR)] and compared using the Mann-Whitney U test or the Kruskal-Wallis test, when more than 2 groups were compared.

**Table 1. Demographic, clinical and laboratory characteristics of participants in the carotid ultrasound study.**

| Variable | CKD patients | Controls | p-values |
|---|---|---|---|
| No. of participants (n) | 200 | 121 | |
| Age (yr) | 64 (54–71) | 64 (52–70) | 0.366 |
| Female sex (n, %) | 76 (38) | 49 (41) | 0.657 |
| BP systolic (mmHg) | 133 (±1) | 130 (±2) | 0.228 |
| BP diastolic (mmHg) | 80 (±1) | 81 (±1) | 0.451 |
| P-creatinine (μmol/l) | 142 (123–165) | 79 (70–89) | <0.001 |
| eGFR (ml/min/1.73m$^2$) | 41 (35–50) | 82 (75–92) | <0.001 |
| P-HDL-cholesterol (mmol/l) | 1.52 (±0.04) | 1.82 (±0.05) | <0.001 |
| P-LDL-cholesterol (mmol/l) | 3.0 (±0.1) | 3.5 (±0.1) | <0.001 |
| P-triglycerides (mmol/l) | 1.55 (1.13–2.27) | 0.88 (0.69–1.15) | <0.001 |
| P-glucose (mmol/l) | 6.3 (±0.1) | 5.8 (±0.0) | 0.006 |
| P-ionized calcium (mmol/l) | 1.22 (±0.00) | 1.23 (±0.00) | 0.548 |
| P-phosphate (mmol/l) | 1.02 (±0.01) | 1.01 (±0.02) | 0.609 |
| Urine albumin/creatinine ratio x 10$^{-3}$ | 67(10–352) | 0 (0–4) | <0.001 |
| P-urate (mmol/l) | 0.44 (±0.01) | 0.31 (±0.01) | <0.001 |
| Hypertension (n, %) | 178 (89) | 35 (29) | <0.001 |
| Hypercholesterolemia (n, %) | 162 (82) | 81 (67) | 0.002 |
| Type 1 diabetes (n, %) | 6 (3) | 0 (0) | <0.001 |
| Type 2 diabetes (n, %) | 42 (21) | 0 (0) | |
| Cardiovascular co-morbidity (n, %) | 41 (21) | 0 (0) | <0.001 |
| Cardiovascular disease and diabetes (n, %) | 18 (9) | 0 (0) | <0.001 |
| Alcohol intake (units/week) | 2 (0–10) | 7 (3–14) | <0.001 |
| Smoking status (n, %): | | | |
| Never smoked | 78 (39) | 63 (52) | 0.028 |
| Former smoker | 82 (41) | 45 (37) | |
| Active smoker | 40 (20) | 13 (11) | |
| Smoking (pack years) | 5 (0–30) | 0 (0–10) | <0.001 |
| Medication (n, %): | | | |
| Lipid lowering | 104 (52) | 0 (0) | <0.001 |
| Anti-hypertensive | 168 (84) | 0 (0) | <0.001 |
| Insulin | 23 (12) | 0 (0) | <0.001 |
| Oral antidiabetic | 23 (12) | 0 (0) | <0.001 |
| Anti-platelet | 62 (31) | 2 (2) | <0.001 |

Values for categorical variables are given as number (percentages), totals may not add up to 100% due to rounding; values for continuous variables are given as mean (±SEM) or median (IQR, 25th to 75th percentile). CKD: chronic kidney disease. BP: blood pressure. eGFR: estimated glomerular filtration rate.

## Results

### Characteristics of the ultrasound study cohort

A total of 200 patients with CKD and 121 controls were enrolled (Table 1).

Among the patients, 21% had type 2 diabetes, 3% type 1 diabetes and 21% cardiovascular disease. A large proportion of the patients were on anti-hypertensive medication (84%), lipid lowering treatment (52%), and/or anti-platelet treatment (31%). Age and sex distribution were similar in patients with CKD and controls (Table 1). Controls had normal eGFR and no albuminuria. Among the controls, 29% had hypertension and 67% hypercholesterolemia

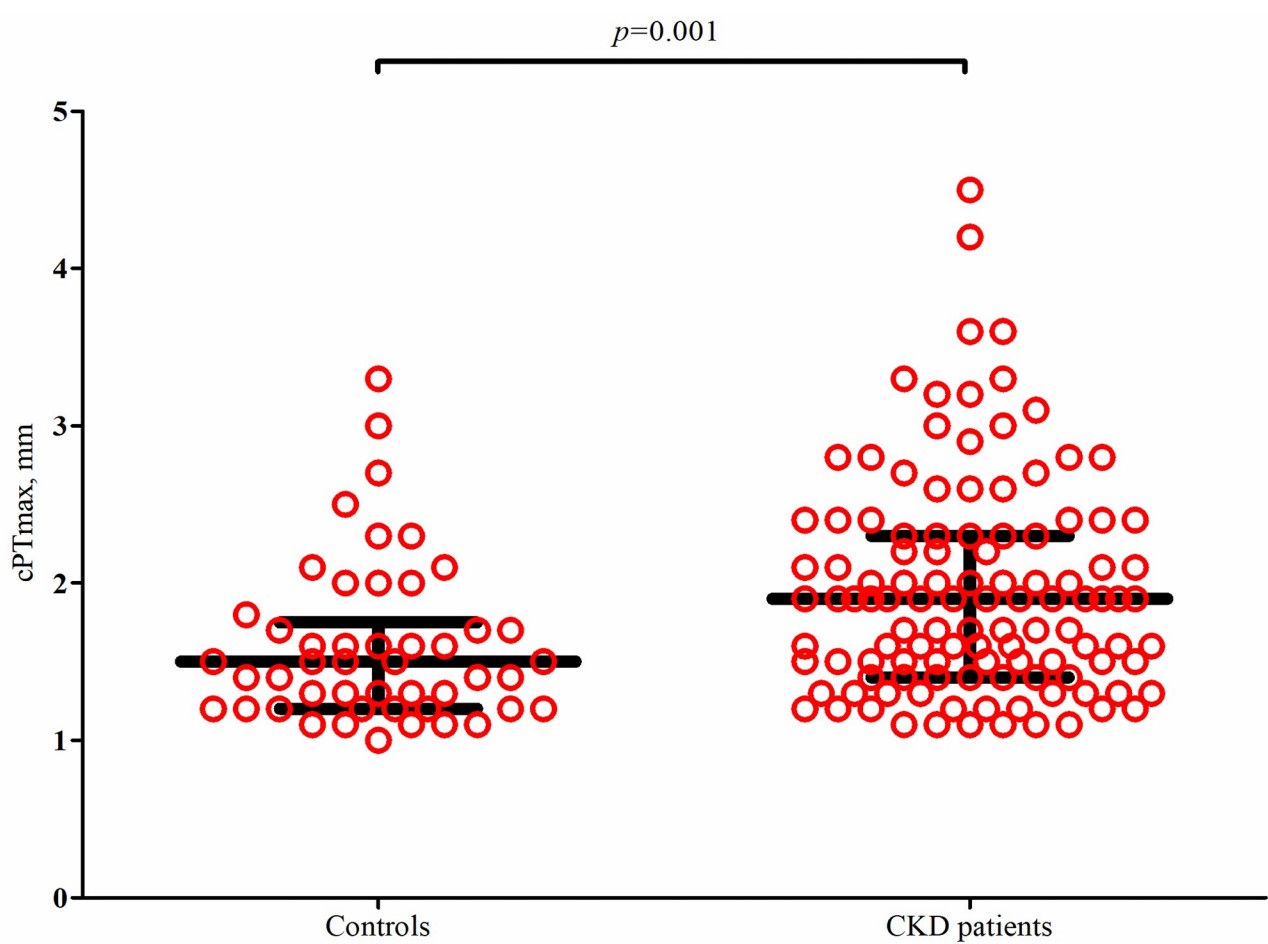

**Fig 2. Distribution of maximal carotid plaque thickness (cPTmax) in patients with CKD stage 3 and controls.** Includes only participants with plaques (115/200 patients and 48/121 controls). Dot plot with median (long black horizonal line) and interquartiles (short black horizontal lines). The *p*-value is from a Mann-Whitney U-test.

diagnosed at inclusion. None received antihypertensive or cholesterol-lowering therapy. Two controls received anti-platelet therapy, but none of them had known cardiovascular disease.

## cPTmax in patients with CKD stage 3 and controls

Plaques were present in 58% of patients with CKD stage 3 (n = 115) as compared to 40% of controls (n = 48), $p$ = 0.002. Among participants with plaques, the median (IQR) cPTmax was significantly higher in patients as compared with the controls (1.9 (1.4–2.3) versus 1.5 (1.2–1.8) mm), $p$ = 0.001 (Fig 2). Comparing patients with plaques but without prevalent cardiovascular disease (n = 82) with controls (n = 48), the difference was still significant (1.7 (1.4–2.2) versus 1.5 (1.2–1.8) mm), $p$ = 0.01.

## Association between cPTmax and prevalent cardiovascular disease

Cardiovascular disease was present in 9% of patients without plaques, 23% of patients with cPTmax 1.0–1.9 mm, and 35% of patients with cPTmax >1.9 mm, $p$ = 0.002 (Table 2). Among patients with plaques, median cPTmax was 2.0 (1.6–2.8) mm in patients with prevalent

**Table 2. Characteristics of patients with CKD grouped according to plaque status.**

| Variable | No plaques | cPTmax 1.0–1.9 mm | cPTmax >1.9 mm | *p*-values (Plaques present vs. absent) |
|---|---|---|---|---|
| No. of participants (n) | 85 | 69 | 46 | |
| Age (yr) | 54 (44–62) | 67 (61–72) | 70 (65–72) | <0.001 |
| Female sex (n, %) | 43 (51) | 26 (38) | 6 (13) | 0.001 |
| BMI (kg/m$^2$) | 28.0 (±0.7) | 29.2 (±0.8) | 29.5 (±0.8) | 0.131 |
| Abdominal circumference (cm) | 96.5 (±2.1) | 104.0 (±2.0) | 107.9 (±2.3) | <0.001 |
| BP systolic (mmHg) | 130 (±2) | 137 (±2) | 134 (±3) | 0.025 |
| BP diastolic (mmHg) | 83 (±1) | 81 (±1) | 76 (±2) | 0.015 |
| P-creatinine (µmol/l) | 138 (120–163) | 139 (120–162) | 151 (126–170) | 0.474 |
| eGFR (ml/min/1.73m$^2$) | 41 (36–52) | 42 (35–50) | 40 (34–44) | 0.175 |
| P-HDL-cholesterol (mmol/l) | 1.50 (±0.06) | 1.63 (±0.08) | 1.36 (±0.07) | 0.780 |
| P-LDL-cholesterol (mmol/l) | 3.2 (±0.1) | 2.9 (±0.1) | 2.6 (±0.1) | 0.003 |
| P-triglycerides (mmol/l) | 1.52 (1.14–2.23) | 1.59 (1.09–2.33) | 1.56 (1.20–2.17) | 0.942 |
| P-glucose (mmol/l) | 5.8 (±0.1) | 6.5 (±0.2) | 6.8 (±0.4) | 0.004 |
| P-ionized calcium (mmol/l) | 1.23 (±0.01) | 1.22 (±0.01) | 1.22 (±0.01) | 0.835 |
| P-phosphate (mmol/l) | 1.03 (±0.02) | 1.02 (±0.02) | 1.04 (± 0.03) | 0.986 |
| Urine albumin/creatinine ratio x 10$^{-3}$ | 70 (11–331) | 47 (7–343) | 125 (31–639) | 0.506 |
| P-urate (mmol/l) | 0.43 (±0.01) | 0.44 (±0.01) | 0.48 (±0.02) | 0.148 |
| Hypertension (n, %) | 74 (87) | 62 (90) | 43 (94) | 0.333 |
| Hypercholesterolemia (n, %) | 65 (77) | 60 (87) | 38 (83) | 0.118 |
| Diabetes (n,%): | | | | 0.001 |
| Type 1 diabetes | 3 (4) | 2 (3) | 1 (2) | |
| Type 2 diabetes | 7 (8) | 20 (29) | 15 (33) | |
| Cardiovascular-comorbidity (n, %) | 8 (9) | 16 (23) | 16 (35) | 0.002 |
| Alcohol intake (units/week) | 1 (0–6) | 4 (0–14) | 3 (1–12) | 0.087 |
| Smoking status (n, %): | | | | 0.010 |
| Never smoked | 42 (49) | 23 (33) | 13 (28) | |
| Former smoker | 33 (39) | 29 (42) | 20 (44) | |
| Active smoker | 10 (12) | 17 (25) | 13 (28) | |
| Smoking (pack years) | 1 (0–15) | 10 (0–38) | 28 (0–50) | <0.001 |
| Medication (n, %): | | | | |
| Lipid lowering | 32 (38) | 41 (59) | 31 (67) | <0.001 |
| Anti-hypertensive | 70 (82) | 56 (81) | 42 (91) | 0.585 |
| Insulin | 6 (7) | 6 (9) | 11 (24) | 0.091 |
| Oral anti-diabetic | 2 (2) | 14 (20) | 7 (15) | <0.001 |
| Anti-platelet | 10 (12) | 25 (36) | 27 (59) | <0.001 |
| cPT max, mm* | NR | 1.5 (1.3–1.7) | 2.4 (2.2–2.9) | NR |
| Coronary calcium score | 0 (0–38) | 76 (0–378) | 507 (87–1012) | <0.001 |
| Carotid calcium score | 0 (0–3) | 88 (10–252) | 356 (153–1104) | <0.001 |

Values for categorical variables are given as number (percentages), totals may not add up to 100% due to rounding; values for continuous variables are given as mean (±SEM) or median (IQR, 25th to 75th percentile). BMI: body mass index. BP: blood pressure. eGFR: estimated glomerular filtration rate. cPTmax: maximal carotid plaque thickness. NR: not relevant.

*cPT max was only measured in patients with plaques.

cardiovascular disease (n = 32) versus 1.7 (1.4–2.2) mm in patients without cardiovascular disease (n = 83, *p* = 0.024). Table 2 demonstrates that patients with carotid plaques had a higher prevalence of multiple classical cardiovascular risk factors as compared with patients without plaques. Patients with diabetes (n = 48) showed a higher prevalence of plaques (79%) than

patients without diabetes (51%, n = 152), $p<0.001$. Nevertheless, among patients with plaques, median cPTmax did not differ between patients with (n = 38) and without (n = 77) diabetes, $p = 0.704$. Likewise, patients with a smoking history showed a higher prevalence of plaques than patients who had never smoked (active, former and never smokers: 75%, 60%, and 46%, respectively), $p = 0.010$. However, among patients with plaques, median cPTmax did not differ between smoking status groups ($p = 0.827$).

## Relation of cPTmax with carotid and coronary artery calcium scores

Fig 3 shows a strong association between cPTmax and carotid as well as coronary artery calcium scores. Carotid and coronary calcium scores >400 were present in 0% and 4%, respectively, of patients with no carotid plaques, in 19% and 24% of patients with cPTmax 1.0–1.9 mm, and in 48% and 53% of patients with cPTmax >1.9 mm, $p<0.001$.

Fig 4 demonstrates that the association between cPTmax and carotid calcium score was maintained, when the CKD patients were grouped according to sex, age or smoking status, i.e. few or no patients with cPTmax >1.9 mm in the calcium score 0 category and few or no patients with no plaques in the calcium score >400 category. The same pattern was also seen when the patients were grouped according to presence/absence of hypertension, hypercholesterolemia and diabetes (S1 Fig). Similar data for the relation between cPTmax and coronary calcium score are shown in S2 and S3 Figs.

For comparison, in the thoracic aorta, abdominal aorta and iliac arteries, calcium scores >400 were present in 20%, 26% and 25%, respectively, of patients with no carotid plaques (Fig 5).

## Discussion

The CPH CKD Cohort study is the first to examine cPTmax determined by ultrasound in patients with CKD. We demonstrated that both plaque prevalence and cPTmax were increased in patients with CKD stage 3 compared with controls. The increase of cPTmax in patients versus controls was maintained, when patients with CKD and prevalent cardiovascular disease were excluded. cPTmax was closely associated with prevalent cardiovascular disease and with calcification of the carotid and coronary arteries.

Increased prevalence of plaques in patients with CKD has been reported by others [26–28]. In the NEFRONA study, 70% of patients with CKD stage 3 had plaques in the carotid or femoral arterial regions compared with 52% of the controls [26]. Since 12% of patients and 10% of controls had plaques in the femoral arteries only, the prevalence of carotid plaques in patients with CKD stage 3 and controls was similar to the findings in our study. The NEFRONA study also reported that progression of atherosclerosis (i.e. number of territories with plaques) was associated with progression of CKD and that progression of atherosclerosis was delayed in patients with no plaques at baseline [29]. Finally, the NEFRONA follow-up study showed that the number of territories with plaques was a stronger predictor of cardiovascular events than the mere presence (versus absence) of plaques in the carotid and femoral arteries of patients with CKD (non-dialysis dependent) [17]. Carotid intima media thickness did not predict cardiovascular events. Another study of 101 patients on hemodialysis demonstrated that the sum of plaque thickness from each segment of the carotid artery on both sides predicted cardiovascular mortality after 4 years of follow-up [18]. These results indicate that quantification of plaque size improves the predictive power in the CKD population.

The presence of plaques in the present study was associated with multiple classical cardiovascular risk factors, and cPTmax was closely associated with prevalent cardiovascular disease. Also, the prevalence of plaques was higher in patients with diabetes, but with similar cPTmax

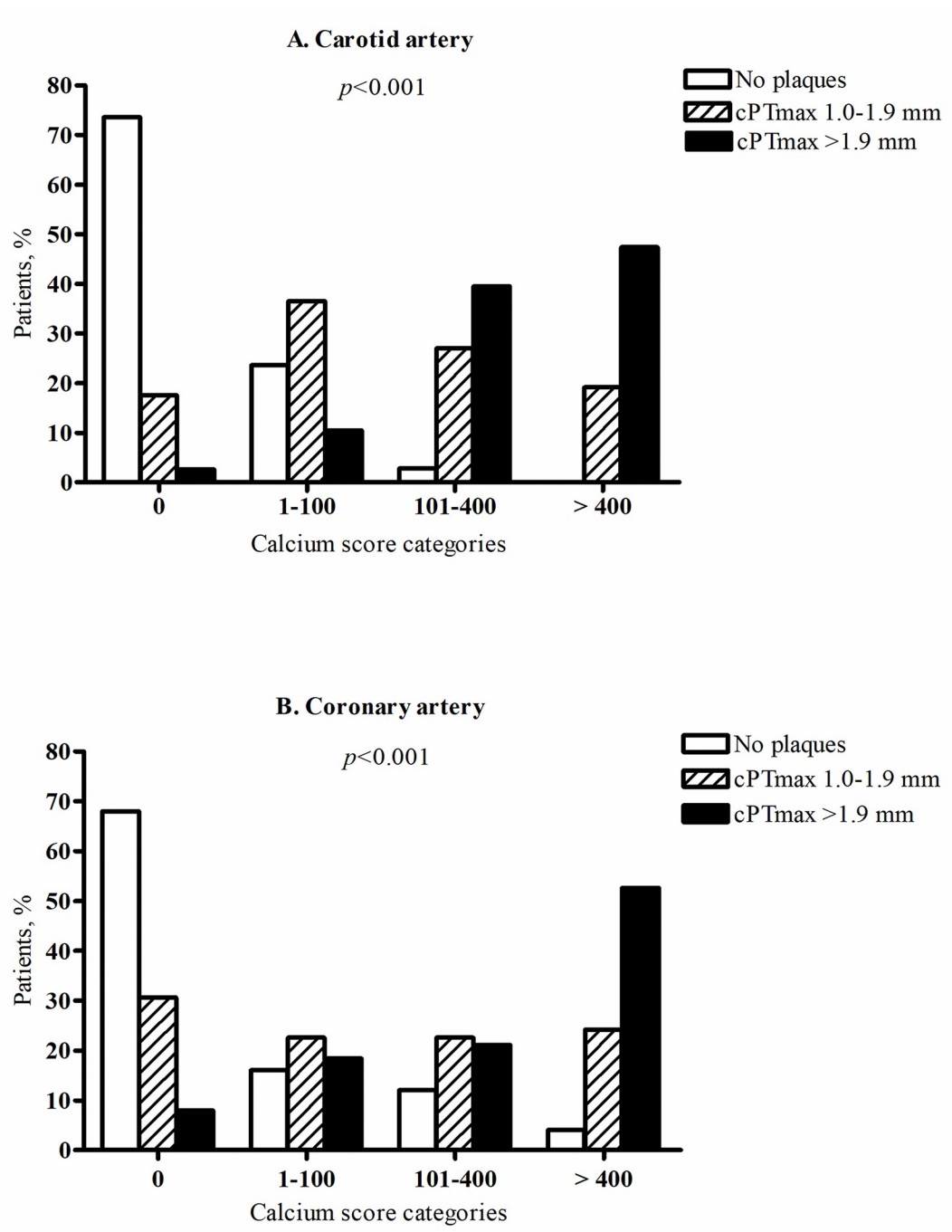

**Fig 3. Association between maximal carotid plaque thickness (cPTmax) and carotid artery calcium score (A) and coronary artery calcium score (B).** According to carotid ultrasound findings, patients were divided into 3 groups: No carotid plaques, cPTmax 1.0–1.9 mm, cPTmax >1.9 mm. Based on the distribution of calcium scores from noncontrast CT scanning of the carotid and coronary arteries, patients were divided into calcium score categories of 0, 1–100, 101–400 and >400. The p-values are from cross tabulation and chi-square analysis (rows: Calcium score categories, columns: cPTmax categories).

A. Men versus women

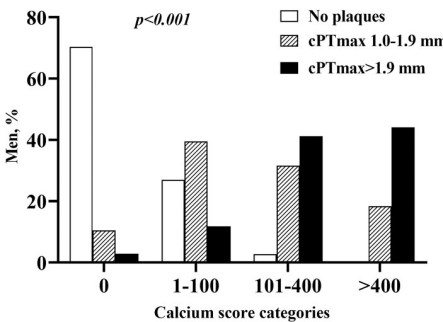
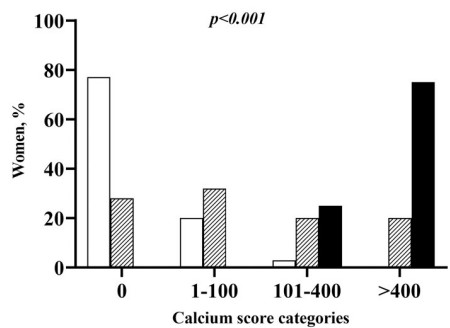

B. Smokers versus non-smokers

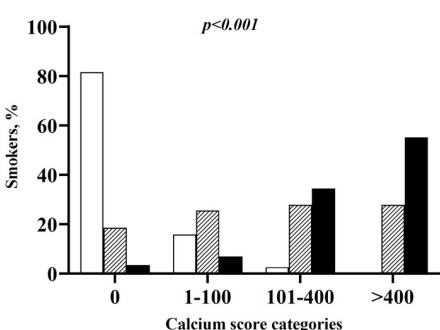
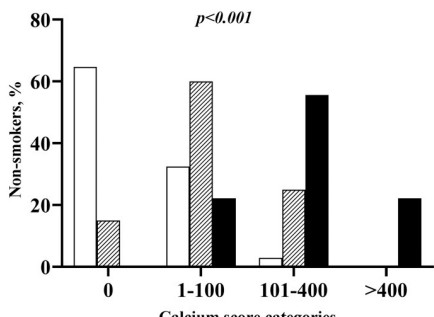

C. Patients aged ≤ 64 yrs versus > 64 yrs

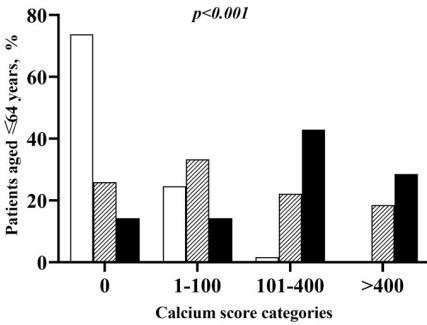
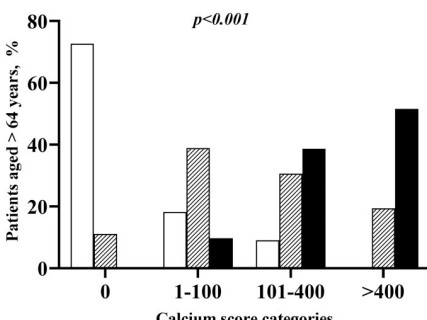

**Fig 4. Association between maximal carotid plaque thickness (cPTmax) and carotid artery calcium score in patients with CKD stage 3.** (A) In men versus women, (B) in smokers versus non-smokers, and (C) in patients aged ≤ 64 yrs. versus > 64 yrs. According to carotid ultrasound findings, patients were divided into 3 groups: No carotid plaques, cPTmax 1.0–1.9 mm, cPTmax >1.9 mm. Based on the distribution of calcium scores from CT scanning of the carotid arteries, patients were divided into calcium score categories of 0, 1–100, 101–400 and >400. Non-smokers are defined as patients with 0 smoking pack yrs. and smokers as patients with smoking pack yrs. > 0. Age 64 yrs. is the median age of the CKD patients. The $p$-values are from cross tabulation and chi-square analysis (rows: Calcium score categories, columns: cPTmax categories).

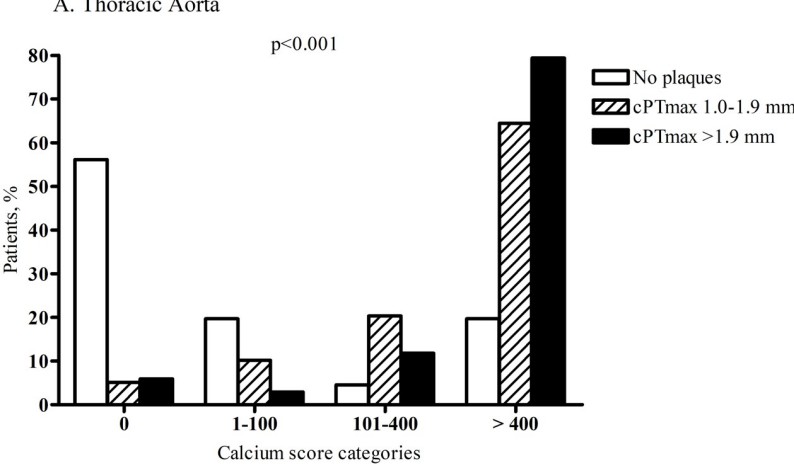

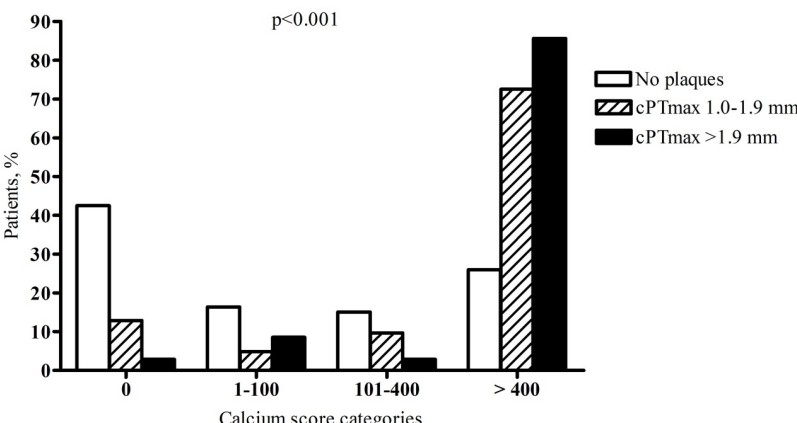

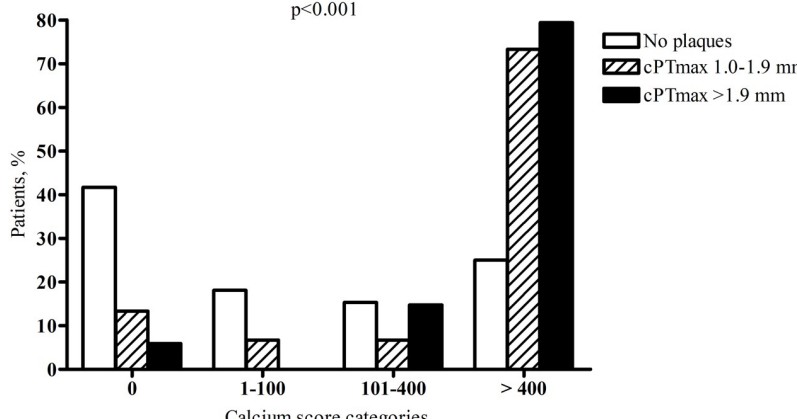

**Fig 5. Association between maximal carotid plaque thickness (cPTmax) and thoracic aortic calcium score (A), abdominal aortic calcium score (B) and iliac artery calcium score (C).** According to carotid ultrasound findings, patients were divided into 3 groups: No carotid plaques, cPTmax 1.0–1.9 mm, cPTmax >1.9 mm. Based on the distribution of calcium scores from noncontrast CT scanning of the thoracic aorta, abdominal aorta and iliac arteries, patients were divided into calcium score categories of 0, 1–100, 101–400 and >400. The *p*-values are from cross tabulation and chi-square analysis (rows: Calcium score categories, columns: cPTmax categories).

in CKD patients with and without diabetes. A possible explanation of the latter could be that the patients had several risk factors in addition to diabetes, including renal failure, and that the number of patients with diabetes was relatively small.

Likewise, patients with a smoking history showed a higher prevalence of plaques than patients who had never smoked, but with a non-significant difference between the median cPTmax of active, former and never smokers. Most likely, this lack of a significant difference is due to the relatively small number of patients with plaques and different smoking status.

The close association between cPTmax and calcium scores in the coronary arteries does indeed support the idea that cPTmax may be usable as a measure of cardiovascular risk in CKD, since CACS seems to improve prediction of cardiovascular risk in CKD patients [13–16]. The association between cPTmax and carotid calcium was very similar to the association with coronary calcium. A small study of hemodialysis patients also showed that carotid plaque thickness correlated strongly with CACS [30], and the authors observed that if there were no plaques, there was also no coronary artery calcification. Although the association between plaque thickness and calcification was strong in our study, we did observe presence of carotid plaques without carotid or coronary calcification in some patients (12 and 22, respectively) and calcification without plaques in others (19 and 24, respectively). This is not surprising, however, since early plaques are not calcified, and since media calcification not related to atherosclerosis is common in CKD (although more common in more advanced CKD). Unfortunately, intimal and medial calcification are indistinguishable by CT scanning. The close association between cPTmax and coronary (and carotid) calcium score suggests that most arterial calcium is localized in the intima of these arteries in patients with CKD stage 3. Interestingly, the association between cPTmax and calcium score categories was markedly less when the thoracic and abdominal aorta and the iliac arteries were considered. Thus, 20–26% of patients showed calcium scores >400 in these arterial regions, even if carotid plaques were absent. In comparison, no patients without carotid plaques showed carotid artery calcium scores >400, and only 4% of these patients showed coronary artery calcium scores >400. This suggests that non-intimal calcification i.e. medial calcification may be more pronounced in the aorta and the iliac arteries. However, since we did not measure PTmax in the femoral arteries, this hypothesis cannot be confirmed or rejected by the available data.

Ultrasound examination of the carotid arteries alone is more convenient and socially acceptable and less time-consuming than ultrasound scanning of both the carotid and femoral arteries. cPTmax is easier to measure than plaque volume, meaning that cPTmax measurements could be performed in a routine ultrasound laboratory. A follow-up study in the CKD Cohort will be needed to confirm the findings from the general population that cPTmax alone predicts cardiovascular risk similarly to CACS.

We recognize that the present study had some limitations. Carotid plaque burden was not measured for comparison with cPTmax. The cross-sectional nature of the study did not allow us to evaluate the predictive value of cPTmax. The main strengths of this study include that it is the first study measuring cPTmax in patients with CKD. We had a large control group. Vascular explorations were performed and evaluated by the same person, thus minimizing measurement variability. cPTmax data were compared with a validated method for cardiovascular risk scoring (CACS).

In conclusion, this is the first study to show that cPTmax is increased in patients with CKD stage 3 and closely associated with prevalent cardiovascular disease and calcification in both the carotid and coronary arteries.

## Supporting information

**S1 Fig. Association between maximal carotid plaque thickness (cPTmax) and carotid artery calcium score in patients with CKD stage 3.** (A) In patients with and without hypertension, (B) in patients with and without hypercholesterolemia, and (C) in patients with and without diabetes. According to carotid ultrasound findings, patients were divided into 3 groups: No carotid plaques, cPTmax 1.0–1.9 mm, cPTmax >1.9 mm. Based on the distribution of calcium scores from CT scanning of the carotid arteries, patients were divided into calcium score categories of 0, 1–100, 101–400 and >400. Hypertension was defined as systolic blood pressure >140 mmHg and/or diastolic blood pressure >90 mmHg or use of oral antihypertensive treatment. Hypercholesterolemia was defined as low-density lipoprotein (LDL) cholesterol >3 mmol/l or treatment with cholesterol-lowering medication. Diabetes: Type 1 or type 2 diabetes. The *p*-values are from cross tabulation and chi-square analysis (rows: Calcium score categories, columns: cPTmax categories).
(TIF)

**S2 Fig. Association between maximal carotid plaque thickness (cPTmax) and coronary artery calcium score in patients with CKD stage 3.** (A) In men versus women, (B) in smokers versus non-smokers, and (C) in patients aged ≤ 64 yrs. versus > 64 yrs. According to carotid ultrasound findings, patients were divided into 3 groups: No carotid plaques, cPTmax 1.0–1.9 mm, cPTmax >1.9 mm. Based on the distribution of calcium scores from CT scanning of the coronary arteries, patients were divided into calcium score categories of 0, 1–100, 101–400 and >400. Non-smokers are defined as patients with 0 smocking pack yrs. and smokers as patients with smoking pack yrs. > 0. Age 64 yrs. is the median age of the CKD patients. The *p*-values are from cross tabulation and chi-square analysis (rows: Calcium score categories, columns: cPTmax categories).
(TIF)

**S3 Fig. Association between maximal carotid plaque thickness (cPTmax) and coronary artery calcium score in patients with CKD stage 3.** (A) In patients with and without hypertension, (B) in patients with and without hypercholesterolemia, and (C) in patients with and without diabetes. According to carotid ultrasound findings, patients were divided into 3 groups: No carotid plaques, cPTmax 1.0–1.9 mm, cPTmax >1.9 mm. Based on the distribution of calcium scores from CT scanning of the coronary arteries, patients were divided into calcium score categories of 0, 1–100, 101–400 and >400. Hypertension was defined as systolic blood pressure >140 mmHg and/or diastolic blood pressure >90 mmHg or use of oral antihypertensive treatment. Hypercholesterolemia was defined as low-density lipoprotein (LDL) cholesterol >3 mmol/l or treatment with cholesterol-lowering medication. Diabetes: Type 1 or type 2 diabetes. The *p*-values are from cross tabulation and chi-square analysis (rows: Calcium score categories, columns: cPTmax categories).
(TIF)

## Acknowledgments

We thank the participants for their invaluable contribution to the study. Freja A. K. Saurbrey is thanked for help with Fig 1.

## Author Contributions

**Conceptualization:** Sasha S. Bjergfelt, Bo Feldt-Rasmussen, Henrik Sillesen, Christina Christoffersen, Susanne Bro.

**Data curation:** Sasha S. Bjergfelt, Ida M. H. Sørensen.

**Formal analysis:** Sasha S. Bjergfelt.

**Funding acquisition:** Bo Feldt-Rasmussen.

**Investigation:** Sasha S. Bjergfelt, Ida M. H. Sørensen, Henrik Ø. Hjortkjær, Nino Landler, Ellen L. F. Ballegaard, Tor Biering-Sørensen.

**Methodology:** Sasha S. Bjergfelt, Klaus F. Kofoed, Theis Lange, Henrik Sillesen.

**Project administration:** Bo Feldt-Rasmussen, Susanne Bro.

**Resources:** Bo Feldt-Rasmussen, Henrik Sillesen.

**Supervision:** Klaus F. Kofoed, Theis Lange, Bo Feldt-Rasmussen, Henrik Sillesen, Christina Christoffersen, Susanne Bro.

**Validation:** Susanne Bro.

**Writing – original draft:** Sasha S. Bjergfelt, Susanne Bro.

**Writing – review & editing:** Ida M. H. Sørensen, Henrik Ø. Hjortkjær, Nino Landler, Ellen L. F. Ballegaard, Tor Biering-Sørensen, Klaus F. Kofoed, Theis Lange, Bo Feldt-Rasmussen, Henrik Sillesen, Christina Christoffersen, Susanne Bro.

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
