## [Decision Letter · Decision Letter 0]

16 Sep 2021

PONE-D-21-10088Carotid plaque thickness is increased in chronic kidney disease and associated with carotid and coronary calcificationPLOS ONE

Dear Dr. Bro,

Thank you for submitting your manuscript to PLOS ONE. After careful consideration, we feel that it has merit but does not fully meet PLOS ONE’s publication criteria as it currently stands. Therefore, we invite you to submit a revised version of the manuscript that addresses the points raised during the review process.

We look forward to receiving your revised manuscript.

Kind regards,

Karin Jandeleit-Dahm

Academic Editor

PLOS ONE

Journal Requirements:

2. Thank you for stating the following in the Competing Interests/Financial Disclosure* (delete as necessary) section:

“I have read the journal´s policy and the authors of this manuscript have the following competing interests:

BFR reports research grants from The NovoNordisk Foundation (Steno Collaborative Grant), HS reports research grants from Philips Ultrasound and Bayer and honoraria from Bayer, Novo Nordisk, Bracco and Philips Ultrasound, TBS reports research grants from Sanofi Pasteur and GE Healthcare, the Lundbeck Foundation and the Novo Nordisk Foundation during the conduct of the study. All other authors: no competing interests.”

We note that one or more of the authors are employed by a commercial company: NovoNordisk Foundation, Philips Ultrasound, Bracco and Bayer

Additional Editor Comments (if provided):

Please amend and clarify according to reviewer's comments

Reviewers' comments:

Reviewer's Responses to Questions

**Comments to the Author**

1. Is the manuscript technically sound, and do the data support the conclusions?

Reviewer #1: Yes

Reviewer #2: Yes

2. Has the statistical analysis been performed appropriately and rigorously? 

Reviewer #1: No

Reviewer #2: Yes

3. Have the authors made all data underlying the findings in their manuscript fully available?

Reviewer #1: Yes

Reviewer #2: Yes

4. Is the manuscript presented in an intelligible fashion and written in standard English?

Reviewer #1: Yes

Reviewer #2: Yes

5. Review Comments to the Author

Reviewer #1: This study has measured cPTmax (max plaque thickness) using ultrasound in patients with CKD. Coronary and carotid calcium scores were also collected. They find that cPTmax is increased in patients with CKD and this is associated with calcium scores. This is a well-written and interesting manuscript. There are some queries around the presentation of the statistical analyses. Some minor changes are recommended.

1) Statistics - p values are include above the graphs in Fig 3. It is unclear where the significance lies and should be included. This information should also be included in all the figure legends (where appropriate). At the moment this is missing. The supplementary figures are also missing the statistical analyses, which should be included.

2) The supplementary data offer some very nice findings that should be moved into the main manuscript. Supp figs 1 and 2 both contain interesting data that would boost the impact of the findings of the paper. I recommend moving them into the main manuscript, complete with statistical analyses. Supp Table 1 would also appropriately fit in the main manuscript.

Reviewer #2: This is a cross sectional baseline study investigating 200 patients with CKD and 121 aged and sex matched controls. The cPTmax was assessed by US and arterial calcification by CT. The cPTmax was increased in patients with CKD and is associated with prevalent CV disease as well as the degree of calcification, of both carotid and coronary arteries.

In general, this is an interesting study.

Comments to be addressed:

• Have results been adjusted for differences in lipids and BP, diabetes, and smoking.

• On page 9, authors compared patients with and without diabetes with higher prevalence of plaques in patients with diabetes but with similar cPTmax. Similar comparisons would be interesting in case of smoking since there is a decent number of patients who were active, never, or former smokers. The comparison between active and never smoked as well as former smokers will be interesting.

• In table1, adding how many had both CV co-morbidity and diabetes will be informative.

• In fig 3, it needs a clear statement what and how statistics have been applied and is P value (P<0.001) is only for comparison among three groups at calcium score category (>400). This needs to be clarified.

• Was the overall amount of plaques (plaque burden) also measured?

• Mean eGFR was 40 ml/min. Was it possible to investigate higher versus lower eGFR with relation to calcification (media versus intima)?

6. PLOS authors have the option to publish the peer review history of their article (what does this mean?). If published, this will include your full peer review and any attached files.

Reviewer #1: No

Reviewer #2: No

---

## [Author Response · Author response to Decision Letter 0]

16 Oct 2021

Point-by-point response to reviewer comments:

Reviewer #1: This study has measured cPTmax (max plaque thickness) using ultrasound in patients with CKD. Coronary and carotid calcium scores were also collected. They find that cPTmax is increased in patients with CKD and this is associated with calcium scores. This is a well-written and interesting manuscript. There are some queries around the presentation of the statistical analyses. Some minor changes are recommended.

1) Statistics - p values are include above the graphs in Fig 3. It is unclear where the significance lies and should be included. This information should also be included in all the figure legends (where appropriate). At the moment this is missing. The supplementary figures are also missing the statistical analyses, which should be included.

Response:

As suggested, we have added information on statistical analysis in legends to figures 2, 3, Supplementary Fig. 1 (now Fig. 4) and Supplementary Fig. 2 (now Fig. 5).

Legend to Fig. 2: “The p-value is from a Mann-Whitney U-test”

Fig. 3: “The p-value is from cross tabulation and chi-square analysis (rows: Calcium score categories; columns: cPTmax categories)”

Fig. 4 (former S1 Fig): “The p-value is from cross tabulation and chi-square analysis (rows: Calcium score categories; columns: cPTmax categories)”

Fig. 5 (former S2 Fig): “The p-value is from cross tabulation and chi-square analysis (rows: Calcium score categories; columns: cPTmax categories)”

2) The supplementary data offer some very nice findings that should be moved into the main manuscript. Supp figs 1 and 2 both contain interesting data that would boost the impact of the findings of the paper. I recommend moving them into the main manuscript, complete with statistical analyses. Supp Table 1 would also appropriately fit in the main manuscript.

Response:

As suggested, we have moved S1 table, S1 Fig and S2 Fig into the main manuscript (new Table 2, new Fig. 4 and new Fig. 5). Information on statistical analysis has been added in the figure legends.

Reviewer #2: This is a cross sectional baseline study investigating 200 patients with CKD and 121 aged and sex matched controls. The cPTmax was assessed by US and arterial calcification by CT. The cPTmax was increased in patients with CKD and is associated with prevalent CV disease as well as the degree of calcification, of both carotid and coronary arteries.

In general, this is an interesting study.

Comments to be addressed:

1) Have results been adjusted for differences in lipids and BP, diabetes, and smoking.

Response:

Adjustments of results for cardiovascular risk factors were complicated by the fact that carotid plaques were absent in several patients and controls. In these subjects, cPTmax could not be measured.

As an alternative, we showed that the association between cPTmax and carotid calcium score was maintained when the CKD patients were grouped according to sex, age or smoking status (S1 Fig. (now Fig. 4))

As suggested, we have also looked into the effect of hypertension, hypercholesterolemia and diabetes (new S1 Fig.). When CKD patients were grouped according to hypertension (present/absent), hypercholesterolemia (present/absent) or diabetes (present/absent), cross tabulation with chi-square statistics showed that the association between carotid calcium score categories and cPT max categories was maintained and significant, except for patients with diabetes (p=0.330). Of note, the numbers in several of the cells of the cross-tables were very small for patients without hypertension, without hypercholesterolemia or with diabetes. This weakens the power of the analysis. 

Similar data for the association between cPTmax and coronary calcium score are shown in new S2 and S3 Figs. 

We have added the following in the result section, page 12, line 215-218 (manuscript with changes marked in red):

“The same pattern was also seen when the patients were grouped according to presence/absence of hypertension, hypercholesterolemia and diabetes (S1 Fig.). 

Similar data for the relation between cPTmax and coronary calcium score are shown in S2 and S3 Figs.”

2) On page 9, authors compared patients with and without diabetes with higher prevalence of plaques in patients with diabetes but with similar cPTmax. Similar comparisons would be interesting in case of smoking since there is a decent number of patients who were active, never, or former smokers. The comparison between active and never smoked as well as former smokers will be interesting.

Response:

We have added the following text in the result section, page 10, line 195-197:

“Likewise, patients with a smoking history showed a higher prevalence of plaques than patients who had never smoked (active, former and never smokers: 75%, 60%, and 46%, respectively), p=0.010. However, among patients with plaques, median cPTmax did not differ between smoking status groups (p= 0.827).”

We have added the following in the discussion section, page 14, line 270-273:

“Likewise, patients with a smoking history showed a higher prevalence of plaques than patients who had never smoked, but with a non-significant difference between the median cPTmax of active, former and never smokers. Most likely, this lack of a significant difference is due to the relatively small number of patients with plaques and different smoking status.”

3) In Table 1, adding how many had both CV co-morbidity and diabetes will be informative.

Response:

As suggested, we have added in Table 1 that 18 patients (9%) had both diabetes and CVD.

4) In fig 3, it needs a clear statement what and how statistics have been applied and is P value (P<0.001) is only for comparison among three groups at calcium score category (>400). This needs to be clarified.

Response:

As suggested, we have added information on statistical analysis in legends to figures.

Legend to Fig. 3: The p-value is from cross tabulation and chi-square analysis (rows: Calcium score categories; columns: cPTmax categories)

Fig. 4 (former S1 Fig): The p-value is from cross tabulation and chi-square analysis (rows: Calcium score categories; columns: cPTmax categories)

Fig. 5 (former S2 Fig): The p-value is from cross tabulation and chi-square analysis (rows: Calcium score categories; columns: cPTmax categories)

5) Was the overall amount of plaques (plaque burden) also measured?

Response:

Only cPT max was measured in the present study. A previous large study (reference no. 12) performed by one of the co-authors of the present paper (HS) showed that cPTmax predicted cardiovascular events similarly to the more comprehensive carotid plaque burden estimates (page 3, line 61-64; page 15, line 299-300).

6) Mean eGFR was 40 ml/min. Was it possible to investigate higher versus lower eGFR with relation to calcification (media versus intima)?

Response:

Our group has previously reported on the association between CKD stage and severity of calcification of the carotid, coronary and other major arteries (reference no. 19). The present study only investigated patients with stage 3. We think that the range of eGFR in the present study was too narrow and the number of participants too small to show an association between eGFR and severity of calcification in the carotid and coronary arteries. 

Unfortunately, intimal and medial calcification are indistinguishable by CT-scanning.

---

## [Decision Letter · Decision Letter 1]

10 Nov 2021

Carotid plaque thickness is increased in chronic kidney disease and associated with carotid and coronary calcification

PONE-D-21-10088R1

Dear Dr. Bro,

We’re pleased to inform you that your manuscript has been judged scientifically suitable for publication and will be formally accepted for publication once it meets all outstanding technical requirements.

Kind regards,

Karin Jandeleit-Dahm

Academic Editor

PLOS ONE

Additional Editor Comments (optional):

The manuscript is now acceptable for publication. No further comments.

Reviewers' comments:

Reviewer's Responses to Questions

**Comments to the Author**

1. If the authors have adequately addressed your comments raised in a previous round of review and you feel that this manuscript is now acceptable for publication, you may indicate that here to bypass the “Comments to the Author” section, enter your conflict of interest statement in the “Confidential to Editor” section, and submit your "Accept" recommendation.

Reviewer #1: All comments have been addressed

Reviewer #2: All comments have been addressed

2. Is the manuscript technically sound, and do the data support the conclusions?

Reviewer #1: Yes

Reviewer #2: Yes

3. Has the statistical analysis been performed appropriately and rigorously? 

Reviewer #1: Yes

Reviewer #2: Yes

4. Have the authors made all data underlying the findings in their manuscript fully available?

Reviewer #1: Yes

Reviewer #2: Yes

5. Is the manuscript presented in an intelligible fashion and written in standard English?

Reviewer #1: Yes

Reviewer #2: Yes

6. Review Comments to the Author

Reviewer #1: The authors have responded appropriately to all of the reviewer comments, which has significantly improved this manuscript.

Reviewer #2: The additional analysis and amended discussion have significantly improved the paper. In particular the expanded statistical analysis has further improved the impact of this manuscript

7. PLOS authors have the option to publish the peer review history of their article (what does this mean?). If published, this will include your full peer review and any attached files.

Reviewer #1: No

Reviewer #2: No

---

## [Editor Report · Acceptance letter]

12 Nov 2021

PONE-D-21-10088R1 

Carotid plaque thickness is increased in chronic kidney disease and associated with carotid and coronary calcification 

Dear Dr. Bro:

I'm pleased to inform you that your manuscript has been deemed suitable for publication in PLOS ONE. Congratulations! Your manuscript is now with our production department. 

Kind regards, 

on behalf of

Professor Karin Jandeleit-Dahm 

Academic Editor

PLOS ONE